# Actin Cytoskeleton Polymerization and Focal Adhesion as Important Factors in the Pathomechanism and Potential Targets of Mucopolysaccharidosis Treatment

**DOI:** 10.3390/cells12131782

**Published:** 2023-07-05

**Authors:** Lidia Gaffke, Estera Rintz, Karolina Pierzynowska, Grzegorz Węgrzyn

**Affiliations:** Department of Molecular Biology, University of Gdansk, Wita Stwosza 59, 80-308 Gdansk, Poland; estera.rintz@ug.edu.pl (E.R.); karolina.pierzynowska@ug.edu.pl (K.P.); grzegorz.wegrzyn@ug.edu.pl (G.W.)

**Keywords:** mucopolysaccharidosis, actin cytoskeleton, polymerization, focal adhesion, therapy

## Abstract

The main approach used in the current therapy of mucopolysaccharidosis (MPS) is to reduce the levels of glycosaminoglycans (GAGs) in cells, the deposits considered to be the main cause of the disease. Previous studies have revealed significant differences in the expression of genes encoding proteins involved in many processes, like those related to actin filaments, in MPS cells. Since the regulation of actin filaments is essential for the intracellular transport of specific molecules, the process which may affect the course of MPSs, the aim of this study was to evaluate the changes that occur in the actin cytoskeleton and focal adhesion in cells derived from patients with this disease, as well as in the MPS I mouse model, and to assess whether they could be potential therapeutic targets for different MPS types. Western-blotting, flow cytometry and transcriptomic analyses were employed to address these issues. The levels of the key proteins involved in the studied processes, before and after specific treatment, were assessed. We have also analyzed transcripts whose levels were significantly altered in MPS cells. We identified genes whose expressions were changed in the majority of MPS types and those with particularly highly altered expression. For the first time, significant changes in the expression of genes involved in the actin cytoskeleton structure/functions were revealed which may be considered as an additional element in the pathogenesis of MPSs. Our results suggest the possibility of using the actin cytoskeleton as a potential target in therapeutic approaches for this disease.

## 1. Introduction

Our awareness of rare diseases is still limited. One of the sizable groups is metabolic disorders, arising from abnormal metabolism and resulting in disruption of basic biochemical processes and alteration of the cell’s internal environment. In the case of mucopolysaccharidoses (MPSs), the substrates deposited in lysosomes are glycosaminoglycans (GAGs), which are long-chain polysaccharides involved in different processes like cell adhesion, molecular signaling and others [1,2]. This is due to the presence of mutations in genes encoding enzymes that are responsible for their degradation [2,3]. The consequence of the occurrence of these mutations is an excessive accumulation of GAGs in various tissues which leads to the dysfunction of many systems, including the nervous and circulatory systems [4]. Reduced joint mobility, short stature, heart disease, skeletal dysfunction and corneal opacity are typical symptoms in affected individuals [4,5,6]. MPSs are genetic disorders inherited in an autosomal recessive manner (all types but MPS II, which is an X-linked disease). There are 13 types of mucopolysaccharidoses, classified on the basis of the absence or malfunction of a particular lysosomal enzyme [2].

For many years, an excessive lysosomal accumulation of GAGs was considered to be the main cause of MPSs [7]. The main approach used so far in therapies is to reduce the level of GAGs in cells, either by restoring the normal enzymatic activities of the proteins responsible for their degradation or by inhibiting their synthesis in cells [8,9,10]. The most commonly used treatments are (i) enzyme replacement therapy (ERT) [11], (ii) hematopoietic stem cell transplantation (HSCT) and (iii) gene therapy (GT) [12]. However, it is known that the available therapies are not completely effective and only provide improvement against some symptoms of the disease [13].

Our recent studies have shown that significant modifications take place in cellular processes and organelles in MPS cells. For example, changes have been observed in apoptosis and autophagy, mitochondrial enzyme levels and the cell cycle [14,15,16,17]. Studies of transcriptomes in all types of MPSs have revealed significant differences in the expression of genes encoding proteins responsible for processes related to actin filaments and microtubules [17]. This may explain the wide variation in symptoms among the different groups of MPS patients and the lack of effectiveness of therapies that rely solely on restoring normal GAG levels. These results have shed new light on the understanding of the pathomechanism of the disease, which until recently has been considered to be considerably less complicated. They have also expanded the range of potential therapeutic approaches.

In non-neural cells, alternate polymerization of the plus-end and depolymerization of the-minus end of microfilaments occur at all times, which require the presence of ATP. Actin filaments enable movements of organelles, as well as those of the entire cell. In addition, they can influence its shape and are part of its external structure. The actin cytoskeleton is essential for the intracellular transport of specific molecules. Another important role of microfilaments is to participate in the formation, strengthening and maturation of dendritic spines, found in neuronal dendrites. This is a part of the excitatory synapse. Dendritic spikes have many receptors and signaling proteins associated with the presynaptic part of the axon. Changes in dendritic spikes occur in many types of neurological disorders [18,19]. Actin-binding proteins (ABPs) are responsible for regulating polymerization processes [20]. Depending on the needs of the cell, microfilaments can form branches. This happens with the help of Arp2/3 proteins which are activated by nuclease-promoting factors: N-WASP and WAVE. Important roles in the regulation of transcription, membrane transport and cell growth are played by Rac and Cdc42 proteins, which belong to the RHO-GTPase family [21]. GTP binding to Rac/Cdc42 activates them [22], and the activation of N-WASP and WAVE occurs in the stimulated cell, induced by the action of Cdc42 [23].

Focal adhesion connects the cytoskeleton to the extracellular matrix (ECM) which surrounds mammalian organs and tissues. The signals that they transmit can regulate adhesions and cell migration but also the processes of apoptosis and gene expression regulation. This affects, among others, development and angiogenesis [24,25,26]. Through integrins, focal adhesion kinase (FAK) is recruited to its complex [27], and activated Vinculin moves and participates in anchoring F-actin to the cell membrane. This protein binds a number of other proteins, including Talin, α-Actinin and Paxillin, which enables the regulation of focal adhesions and embryonic development [28,29].

During changes in the structure of synapses that occur during learning or aging, the formation of dendritic spines is also modified. The same happens when a neurodegenerative disease, such as dementia or epilepsy, occurs [30,31]. Actin microfilaments and their associated proteins are responsible for the proper formation of dendritic spines [32]. The Arp2/3 protein complex is responsible for the polymerization of filaments, participating in the formation of the nerve cell skeleton [33]. Other molecules, containing a domain from the WASP protein homolog, are also involved in microfilament nucleation [34,35]. An important role is also played by Profilin which, by linking actin monomers, is responsible for modifying spike structures, providing synaptic plasticity [36]. Synaptic activity significantly affects the reorganization of the actin cytoskeleton [37]. The disruption of these mechanisms is one of the hallmarks of neurological diseases. For example, in several types of MPSs, increased levels of the expression of genes encoding proteins that attach to the plus-end of microfilaments, regulating the structure of the actin cytoskeleton, have been found [17]. It is also known that abnormalities in the organization of the cytoskeleton occur in the disappearance of synaptic connections in Alzheimer’s disease patients [38]. 

Given how important actin filaments are in many ways, the proteins that regulate their structure and function could be potential therapeutic targets for the treatment of many conditions, including MPSs. On the other hand, this problem has not been comprehensively addressed to date. Therefore, the aim of this study was to assess changes occurring in the actin cytoskeleton and focal adhesion in MPSs, and to estimate whether they might be potential therapeutical targets in the treatment of different types of this disease.

## 2. Materials and Methods

### 2.1. Cell Lines 

MPS types I (p.Trp402Ter/p.Trp402Ter in the *IDUA* gene) and II (p.His70ProfsTer29/- in the *IDS* gene) and the control cell line (control) were employed. All lines were from the Coriell Institute for Medical Research (Camden, NJ, USA) and cultured under standard conditions. Other data for RNA-seq study were deposited with the PRJNA562649 (Sequence Read Archive, SRA) number and methods have been described previously [17]. A complete list of cell lines used for transcriptomic analysis is shown in Appendix A.

### 2.2. Mouse Model 

Mice (the *Idua*^−/−^ line) were obtained from The Jackson Laboratory (Bar Harbor, ME, USA; B6.129-Iduatm1Clk/J; #004068) and were bred with constant access to water and food under standard conditions. At the age of 6 months (N = 6 for each group) they were sacrificed, and brains and livers were collected. All experiments were carried out in accordance with the guidelines of the Council of the European Communities (2010/63/EU) after approval by the Local Ethics Committee for Animal Experiments (Bydgoszcz, Poland) (decision no. 13/2020).

### 2.3. Reagents 

Aldurazyme (laronidase, recombinant human α-L-iduronidase; Genzyme, Sanofi Co., Amsterdam, The Netherlands) and Elaprase (idursulfase, recombinant human 2-iduronate sulfatase; Takeda Ltd., Tokyo, Japan) were used at final concentrations of 0.58 mg/mL and 0.5 mg/mL, respectively. The enzymes were gifts from the Institute “Monument—Child Health Center” (Warsaw, Poland). Genistein was purchased from the Pharmaceutical Research Institute in Poland and added for 24 h at 50 µM concentration.

### 2.4. F-Actin Staining

On coverslips, 5 × 10^4^ cells were cultured overnight. After incubation for 24 h in the presence of appropriate enzymes, genistein, or its vehicle, CellMask™ Green Actin Tracking Stain (Thermo Scientific, Waltham, MA, USA), was added according to the manufacturer’s instructions. The fixation with 2% paraformaldehyde was performed, followed by washing with 0.1% Triton X-100. Another washing procedure was applied 5 times with PBS, and then coverslips were mounted on slides. They were analyzed under a fluorescence microscope (Leica, Wetzlar, Germany). The intensity of immunofluorescence signal was quantified using the software included with the microscope (Leica LAS AF Lite 4.0). After normalizing the obtained data, the relevant graphs of relative fluorescence intensity were prepared and statistical analysis (ANOVA) was performed (GraphPad Prism 9).

### 2.5. Flow Cytometry

Throughout the 10 cm dish plates, 5 × 10^5^ cells were spread evenly and left overnight. After incubation for 24 h in the presence of appropriate enzymes, genistein, or its vehicle, CellMask™ Green Actin Tracking Stain (Thermo Scientific, Waltham, MA, USA), was added according to the manufacturer’s instructions. Samples were collected through trypsinization and centrifugation processes (4 washes in cell staining buffer (BioLegend, San Diego, CA, USA) and analyzed and quantified via an Amnis FlowSight Imaging Flow Cytometer, Luminex (Austin, TX, USA), using IDEAS 6.2 Software. After normalizing the obtained data, the relevant graphs of relative fluorescence intensity were prepared and statistical analysis (ANOVA) was performed (GraphPad Prism 9).

### 2.6. Western Blotting Analysis 

In cellular experiments, 5 × 10^5^ cells were spread evenly throughout the 10 cm dish plates and left overnight. After incubation for 24 h in the presence of appropriate enzymes, genistein, or its vehicle, cells were collected through centrifugation and were lysed for 30 min (1% Triton X-100, 0.5 mM EDTA, 150 mM NaCl, 50 mM Tris, pH 7.5, with protease and phosphatase inhibitors). After another centrifugation step (13,000 rpm, 10 min, 4 °C), supernatants were kept for further analysis. In the animal tissue experiments, brains and livers were flooded with liquid nitrogen, and after pulverization, the resulting powder was weighed and added at a ratio of 1:20 T-PER Tissue Protein Extraction Reagent (Thermo Scientific, Waltham, MA, USA) with protease inhibitors. The samples were then centrifuged (13,000 rpm in a microcentrifuge) for 15 min. The obtained supernatants (protein lysates) were transferred to new tubes. Automatic Western-blotting procedures were employed (ProteinSimple, San Jose, CA, USA) using the 12–230 kDa separation module (ProteinSimple, San Jose, CA, USA). As primary antibodies, those from Actin Nucleation and Polymerization Antibody Sampler (Cell Signaling Technology, Boston, MA, USA); Focal Adhesion Protein Antibody Sampler Kit (Cell Signaling Technology, Boston, MA, USA) and anti-Sec24A (Cell Signaling Technology, Boston, MA, USA) were used, all diluted 1:50. Secondary anti-rabbit antibody from the anti-rabbit detection module (ProteinSimple, San Jose, CA, USA) was used for signal detection. Protein levels were determined via WES, using the total protein chemiluminescence detection module (ProteinSimple, San Jose, CA, USA) as a loading control.

## 3. Results

Recent studies on different gene expression patterns in cells derived from MPS patients have suggested that the dysregulation of genetic control might explain a lack of complete effectiveness of therapies that rely on restoring normal GAG levels [17]. Taking into consideration how important actin filaments are in many aspects, one might propose that proteins that regulate their structure and function could be potential therapeutic targets for the treatment of mucopolysaccharidoses. 

### 3.1. Influence of Enzyme Replacement Therapy and Genistein on F-Actin Level in MPSI/II Cells

Experiments conducted with fibroblasts taken from patients with MPS types I and II indicated a significant increase in F-actin levels compared to controls (Figure 1). For MPS II, both of the tested therapies had a similar, moderately positive effect. However, in the case of MPS I, genistein lowered the protein levels more effectively than the enzyme (Figure 1). These results were also confirmed via flow cytometry (Appendix A). The observed differences in the cell morphology between micrographs in Figure 1 and Appendix A are due to the need to collect cells for flow cytometry analysis. The use of trypsin affected the cell shape and it can explain the lack of microfilaments in fibroblasts, presented in Appendix A, despite using the same dye as in Figure 1.

### 3.2. Levels of Proteins Engaged in Actin Nucleation and Polymerization Processes

The baseline protein levels in patients with MPS I or II were assessed. For both lines, elevated Profilin levels were observed. In contrast, decreased protein levels were observed for Rac1/Cdc42, its phosphorylated form, and Arp3, compared to controls. The levels of Arp2 protein in the patients were within the range of those in the control. The most interesting results were observed for N-WASP and WAVE-2, which indicated trends opposite to those in control cells, namely decreased in MPS I and increased in MPS II (Figure 2). 

Analogous analyses were performed in selected organs of mice with MPS I (Figure 3). Elevated levels of Profilin and reduced levels of N-WASP, WAVE-2 and ARP3 in the brain were confirmed. Interestingly, there were elevated levels of pRac1/Cdc42 in both of the organs tested and reduced levels of Rac1/Cdc42 in the brain. The only statistically significant change involving the liver as the only altered organ was a reduction in ARP2 levels compared to the healthy mouse.

Next, we investigated whether a reduction in GAGs in MPS cells treated with Aldurazyme (MPS I), Elaprase (MPS II) or genistein (an SRT-based treatment) could correct observed changes in the levels of the tested proteins (Figure 2). The most prominent change was noted for Profilin. The protein levels decreased with all therapies for all MPS types tested. The changes were also reversed after using all therapies for WAVE-2 and N-WASP proteins, but only for MPS II. In contrast, apart from Profilin, only N-WASP levels were corrected via Aldurazyme treatment.

### 3.3. Levels of Proteins Engaged in Focal Adhesion

Reduced levels of α-Actinin and Talin-1 were noted, as were elevated levels of Paxillin, Tensin 2 and Vinculin in cells derived from MPS I and II patients, relative to controls. Opposite trends of changes were observed for the FAK protein, where decreased levels in MPS I and increased levels in MPS II were observed (Figure 4).

Analogous analyses were performed in the selected organs of MPS I mice (Figure 5). The levels of all of the tested proteins were reduced, relative to controls. For FAK, Tensin-2 and Vinculin, this was true for both organs tested. For the remaining proteins (Paxillin, α-Actinin and Talin-1), changes were noted only in the liver.

Next, we investigated whether a reduction in GAG storage in MPS cells treated with Aldurazyme (MPS I), Elaprase (MPS II) or genistein (an SRT-based treatment) could correct observed changes in the levels of the tested proteins (Figure 4). The applied therapies proved to be effective, especially for MPS II, where the levels of Talin-1, Vinculin and Tensin 2 were corrected by the enzyme. An apparent effect of both enzyme and genistein was observed for Paxillin, for MPS I and II. Both of the tested therapies proved to be effective in restoring the correct FAK protein levels in MPS II. The most effective therapy was noted for *α*-Actinin, where all forms of the treatment for all of the tested MPS types showed similar efficacy in restoring the originally reduced protein levels.

### 3.4. Transcriptomic Analysis of Changes in the Processes Involving Actin Cytoskeleton in MPS Cells

In order to determine an abundance of changes in the expression of genes coding for the proteins involved in the actin-filament-based process (GO:0030029: QuickGO database of the Gene Ontology Consortium (http://geneontology.org/, accessed on 5 May 2023) in 11 types of MPSs, we analyzed a number of transcripts whose levels were significantly altered in MPS fibroblasts in comparison to healthy cells. The highest number of changed transcripts (48) occurred in MPS IVB, while the lowest (16) occurred in MPS VI. In each of the studied MPS types, we found both transcripts whose expression levels were significantly reduced and those where they were excessively high. However, in all cases, there was a preponderance of an increased expression (Appendix A).

To investigate whether there were transcripts common to most types of MPSs with expression levels different from normal, we looked for transcripts whose levels were altered in at least 5 (out of 11 investigated) types of the disease. Thirteen such cases related to actin processing were found, as listed in Table 1. Four of them were down-regulated, while the rest were up-regulated (among them, three transcripts represented the same gene, *CAPG*).

We also found that the levels of some transcripts were particularly strongly altered in MPS cells. Considering the logarithm value of the fold change (log_2_FC), we observed that the expression of nine genes was up-regulated or down-regulated (at *p* < 0.1) by more than sixteen times (log_2_FC > 4 or log_2_FC < −4) (Figure 6).

The genes from the ‘actin filament-based process’ GO term, whose expressions were changed in MPS cells, are depicted in the volcano plot, shown in Figure 7. Among them, there are down-regulated genes coding for caveolin (*CAV1*) and Rho Family GTPase 3 (*RND3*), a member of the small GTPase protein superfamily. Reduced levels of caveolin in MPS II have been documented by us previously [39].

On the other hand, examples of up-regulated genes were those coding for profilin (*PFN1*), which plays an important role in actin dynamics by regulating actin polymerization in response to extracellular signals, and capping actin protein (*CAPG*), which contributes to the control of actin-based motility in non-muscle cells. What is important is that both genes were up-regulated in almost all MPS lines among the 11 tested. We also supported the transcriptomic data by examining protein levels. We determined the elevated Profilin levels not only in the MPS I mouse liver (Figure 3) but also in the cells of patients with MPS I and MPS II, and evaluated the therapeutic potential of the respective enzymes and genistein in this aspect (Figure 2). Moreover, we confirmed the elevated levels of CAPG [40] and Sec24A (a component of coat protein II (COPII)-coated vesicles that mediate protein transport from the endoplasmic reticulum) in MPS I (Figure 8). Importantly, the effects observed after treatment with the recombinant enzymes (decreased protein levels) indicated the potency of the latter one in designing therapies in the future, at least to some extent (Figure 8).

In addition, the transcriptomic analyses were visualized using the KEGG pathway database ‘Regulation of actin cytoskeleton’ term (Figure 9). This visualization indicates changes in specific processes that are either up- or down-regulated in MPSs.

## 4. Discussion

The main finding described in this report indicates the relevance of actin cytoskeleton elements in the pathomechanism of MPSs. Significant changes were observed in the expression of genes and levels of proteins involved in the actin cytoskeleton processing and focal adhesion. The analysis of the KEGG pathway presenting the ‘regulation of actin cytoskeleton’ process indicated the specific dysregulations occurring in MPS cells. Importantly, these results were corroborated by studies with the MPS I mouse model, where different organs (the brain and liver) were analyzed. The observed defects could be partially corrected by either the application of the active form of dysfunctional enzyme (ERT) or a reduction in the synthesis of GAGs due to treatment with genistein (SRT). This demonstrated that actin cytoskeleton polymerization and focal adhesion may be considered as potential targets for drugs developed to treat MPSs. On the other hand, only partial improvement via ERT or SRT suggests that novel therapeutic approaches focused on the correction of the actin cytoskeleton and focal adhesion are desirable, which might be used either alone or (more probably) together with other therapies, like those mentioned above.

When searching for novel therapies for MPSs, which should be focused on the actin cytoskeleton and focal adhesion, it is important to analyze similar changes occurring in them in other diseases. The regulation of actin filaments is essential for all cells, but is particularly critical for neurons regulating neuronal growth, organelle biogenesis, axon stability and synaptic function [41,42,43,44]. Abnormalities in the actin cytoskeleton have been linked to many diseases, e.g., amyotrophic lateral sclerosis [45], Friedreich’s ataxia [46], Huntington’s disease (HD) [47,48], Alzheimer’s disease [49] and even prion disease [50], and they can play a role in immune response dysregulation [51]. Hirano bodies can be found in brain patients suffering from chronic alcoholism and in the aging of the neurons [52,53]. It has been proposed that using compounds like tetracycline, oxytetracycline, doxycycline and minocycline can be successive against rod-like structures and be used as therapeutic options for different neurodegenerative diseases, where Hirano body formation occurs [54].

Among the more than sixty identified genes related to amyotrophic lateral sclerosis (ALS), there are eight encoded key cytoskeletal proteins, including ALS2 and PFN1. These proteins regulate actin polymerization which is associated with early endosome dynamics [45]. Other studies have indicated the importance of the actin cytoskeleton for spinal muscular atrophy (SMA), where the reduced expression of genes coding for actin cytoskeleton regulators occurs [55]. One study pointed to potential therapeutic targets. Indeed, the modulation of actin homeostasis could abolish nuclear pore instability and dysfunction caused by a *PFN1* mutation [56].

Although aggregates of mutant huntingtin appear to be major causes of Huntington’s disease (HD), its role remains unclear. Over 100 interactions between huntingtin and different proteins are known [57]. The abnormal morphology of the HD fibroblast is correlated with actin cap deficiency, and it was suggested that this unique feature can be used as a personalized biomarker of HD [58]. The Rho-Rac GTPase cascade was identified to be involved in the reorganization of the cytoskeleton by actin-binding proteins cofilin and profilin, and it was concluded that the loss of phosphoSer138 in profilin is correlated with the symptomatic course of HD [59].

Tumor cells are characterized by highly invasive and enhanced actin polymerization activity and an aberrant expression of actin-regulating proteins [60,61]. It was demonstrated that the tumor suppressor adenomatous polyposis coli-dependent actin contributes to maintaining F-actin levels, as well as E-cadherin and Occludin at cell junctions [62]. Actin cytoskeleton is used as a therapeutic target in tumor motility through the pharmacological inhibition of actin assembly. It can be achieved via the inhibition of the Arp2/3 complex or drugs affecting nucleation promotion [63]. Other results have indicated the importance of actin cytoskeleton dynamics and activity of Rho GTPases as targets for the reverse resistance of glioblastoma tumors with wild-type p53 protein [64].

Previous studies on actin cytoskeleton polymerization and focal adhesion in MPSs are scarce. Nevertheless, the importance of focal adhesion for MPSs has been indicated, as cell polarization and migration defects have been found to contribute to Sanfilippo disease (MPS type III) [65]. MPS IIIB mice were used as models, and it has been suggested that soluble extracellular heparan sulfate can enhance focal adhesion formation. Such an effect has been associated with the activation of focal adhesion kinase (FAK) [65]. Importantly, another study provided evidence for the effect of genistein on cervical cancer cells, mediated through the focal adhesion pathway [66]. This may suggest that the genistein-mediated correction of biochemical, organellar and behavioral defects in MPS cellular and mouse models, as described previously [67,68,69,70,71], could arise not only from the inhibition of GAG synthesis, as proposed earlier [72], but also from the effects of this compound on the actin cytoskeleton and focal adhesion. Such a suggestion can also be corroborated by the results presented in this work, where genistein improved defects in both of them. Indeed, as we pointed out earlier, under the influence of enzyme therapy (for both MPS I and MPS II), there was a reduction in the number of altered lysosomes, which made the cells virtually indistinguishable from the healthy ones. Similarly, improvements were obtained in mitochondrial length for MPS II, and in the case of MPS I cells, the treatment with α-L-iduronidase normalized the average coverage by the endoplasmic reticulum [40]. It turns out that correcting GAG storage with enzymes can improve the levels of proteins that were originally altered. This assumption was corroborated through measuring the levels of the Sec24A protein (Figure 9). Such a scenario can also be true in the case of genistein treatment. The mechanism of action of genistein, on the one hand, is based on a reduction in substrate synthesis, but on the other hand, this compound induces the autophagy process which is currently explored in many diseases as a therapeutic strategy. GAGs play many important functions in cells and tissues, being essential components. Therefore, any lowering of their levels must be balanced. The same should be true for designing new therapeutic targets focused on essential proteins. In fact, genes related to the actin cytoskeleton belong to the housekeeping genes; however, we would like to emphasize that any possible therapies should result in the alleviation of elevated/reduced levels of specific proteins, rather than creating genetic knockouts or total silencing of their expression. It is possible that a therapy that takes into account the elimination of GAG storage along with effects on additional elements, such as targeting disruption of the actin cytoskeleton, would be the most effective.

## 5. Conclusions 

The expressions of genes coding from proteins related to the actin cytoskeleton and focal adhesion, as well as the levels of the corresponding protein, are significantly changed in MPS cells and the tissues of MPS I mice. These findings support the proposal that actin cytoskeleton polymerization and focal adhesion are both important factors in the pathomechanism and potential subsidiary targets for the treatment of MPSs.

## Figures and Tables

**Figure 1 cells-12-01782-f001:**
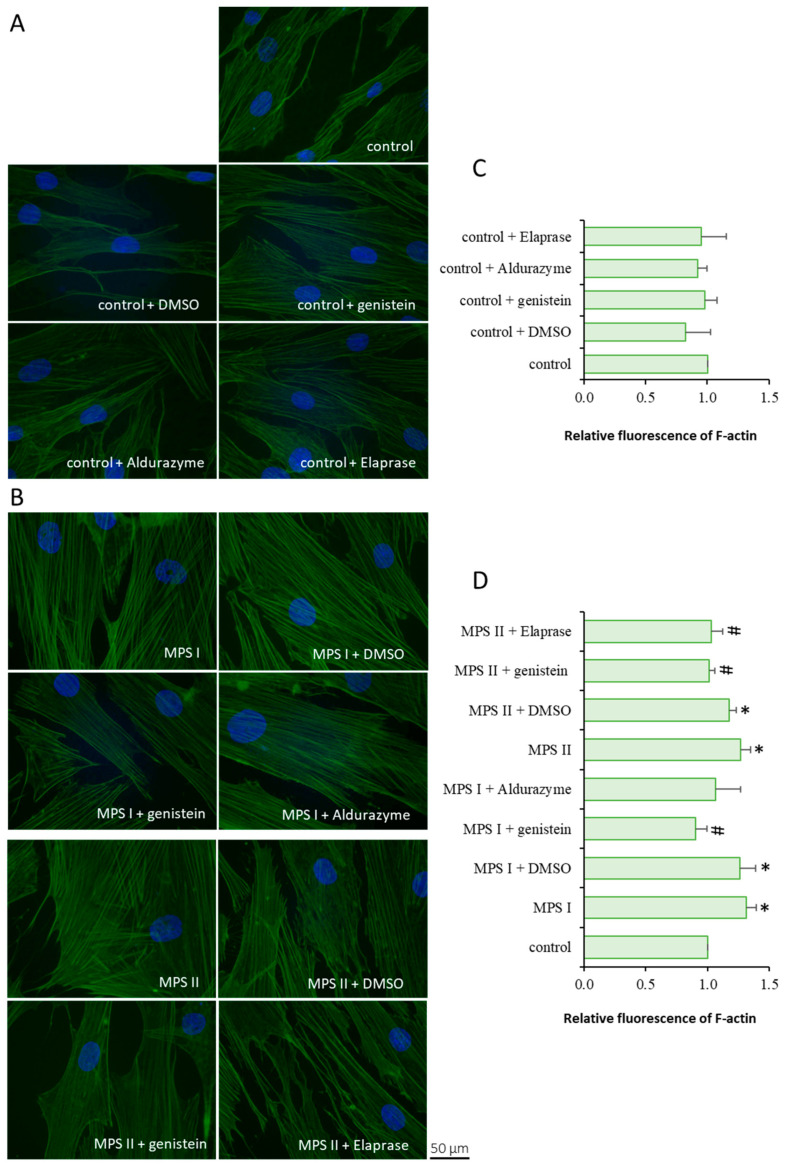
F-actin abundance in cells incubated in the presence of genistein or appropriate enzyme. Investigated lines of fibroblasts were incubated for 24 h and stained with CellMask™ Green Actin Tracking Stain. Panels (**A**,**B**) demonstrate representative micrographs of treated control lines and MPS cells, respectively. Panels (**C**,**D**) show quantitative analysis from 100 randomly chosen cells. Error bars represent standard deviation of three independent repetitions of a given experiment. Statistical analysis was performed using one-way ANOVA test. Differences were considered to be statistically significant relative to the control cell line (*) or untreated MPS I or MPS II (#) when *p* < 0.05.

**Figure 2 cells-12-01782-f002:**
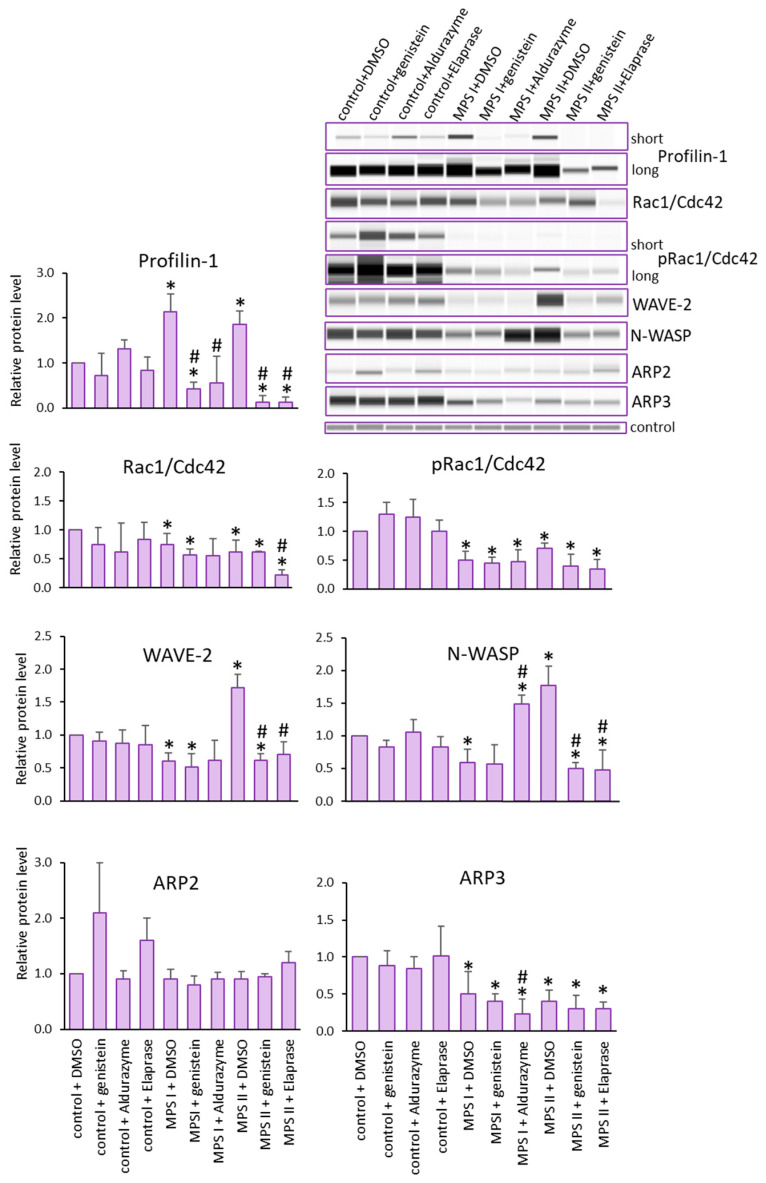
Levels of proteins detected using Actin Nucleation and Polymerization kit. Investigated lines of fibroblasts were incubated in the presence of genistein or appropriate enzyme. Relative levels of the proteins were measured using the Western-blotting procedure. Representative blots are shown and data were quantified via densitometry. Error bars represent standard deviation of three independent repetitions of a given experiment. Differences were considered to be statistically significant relative to the control cell line (*) or untreated MPS I or MPS II (#) when *p* < 0.05.

**Figure 3 cells-12-01782-f003:**
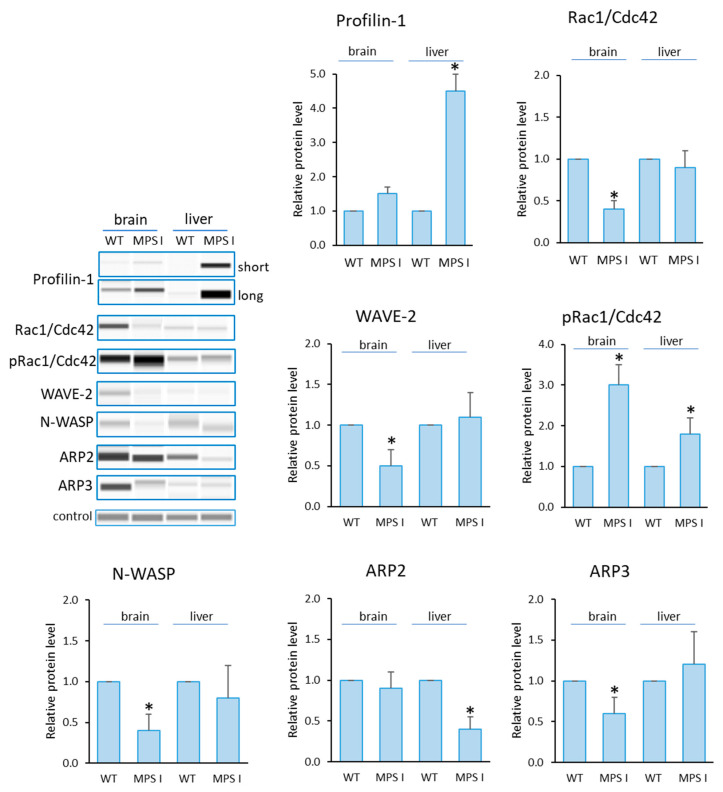
Levels of proteins detected using Actin Nucleation and Polymerization kit in MPS I mice. Relative levels of the proteins were measured using the Western-blotting procedure. Representative blots are shown and data were quantitated via densitometry. Error bars represent the standard deviation of six individuals of a given group. Differences were considered to be statistically significant relative to the wild-type (WT) mice (*) when *p* < 0.05.

**Figure 4 cells-12-01782-f004:**
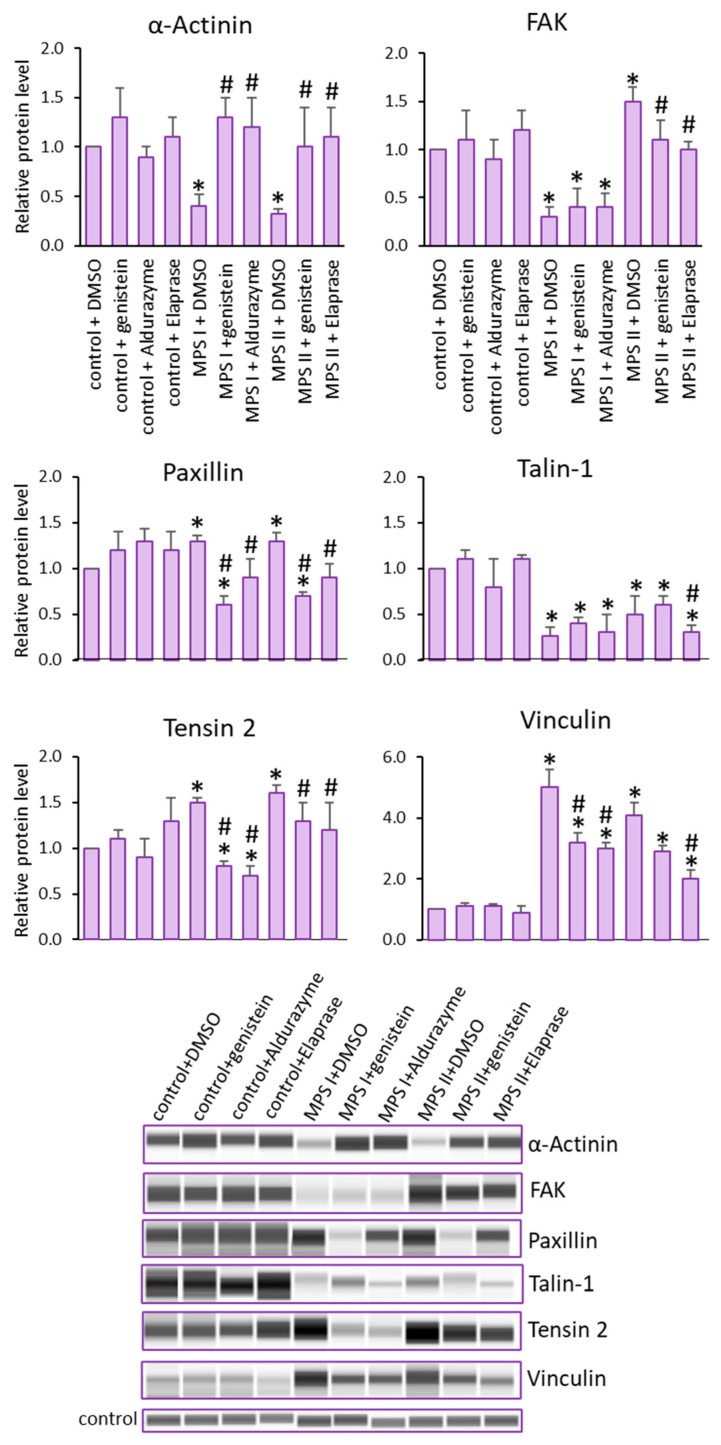
Levels of proteins detected using the Focal Adhesion kit. Investigated lines of fibroblasts were incubated in the presence of genistein or appropriate enzyme. Relative levels of the proteins were measured using the Western-blotting procedure. Representative blots are shown and data were quantitated via densitometry. Error bars represent standard deviation of three independent repetitions of a given experiment. Differences were considered to be statistically significant relative to the control cell line (*) or untreated MPS I or MPS II (#) when *p* < 0.05.

**Figure 5 cells-12-01782-f005:**
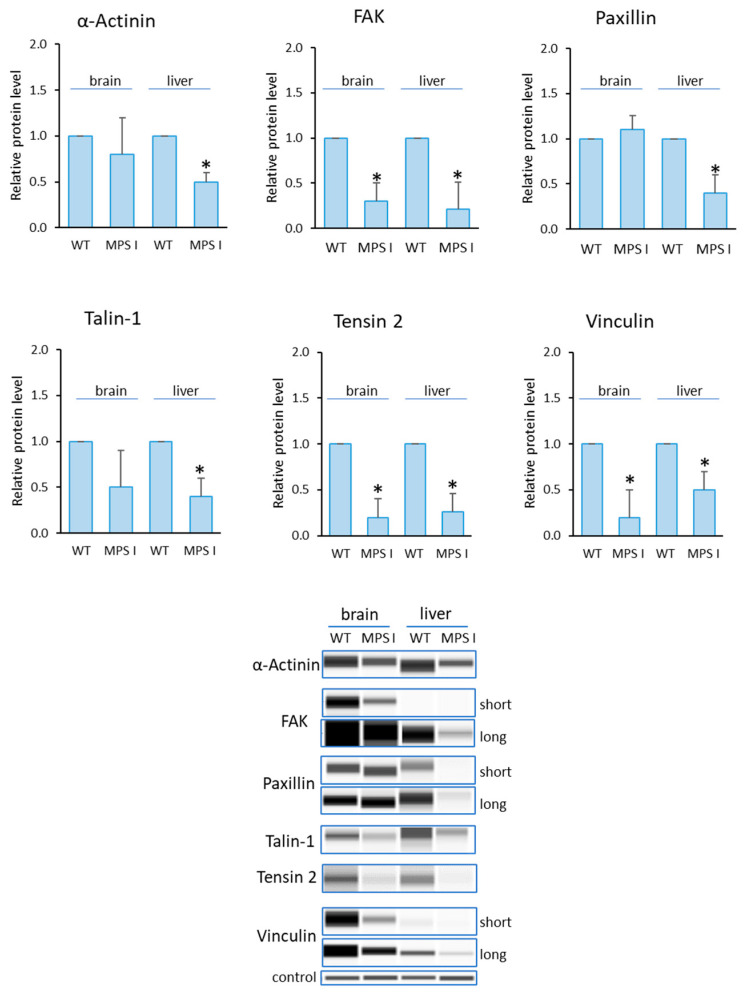
Levels of proteins detected using the Focal Adhesion kit in MPS I mice. Relative levels of the proteins were measured using the Western-blotting procedure. Representative blots are shown and data were quantified via densitometry. Error bars represent standard deviation of six individuals of a given group. Differences were considered to be statistically significant relative to the wild-type (WT) mice (*) when *p* < 0.05.

**Figure 6 cells-12-01782-f006:**
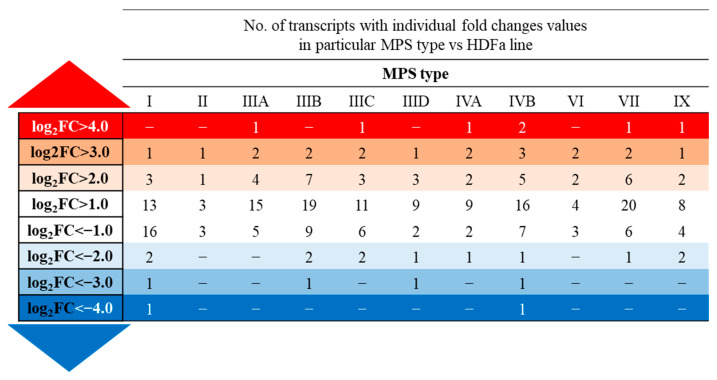
Number of transcripts included in actin-filament-based process (GO:0030029) with altered expression (*p* < 0.1 in one-way ANOVA and post hoc Student’s *t*-test with Bonferroni correction) depending on the level of fold change (FC) value in different types of MPS relative to HDFa control cells.

**Figure 7 cells-12-01782-f007:**
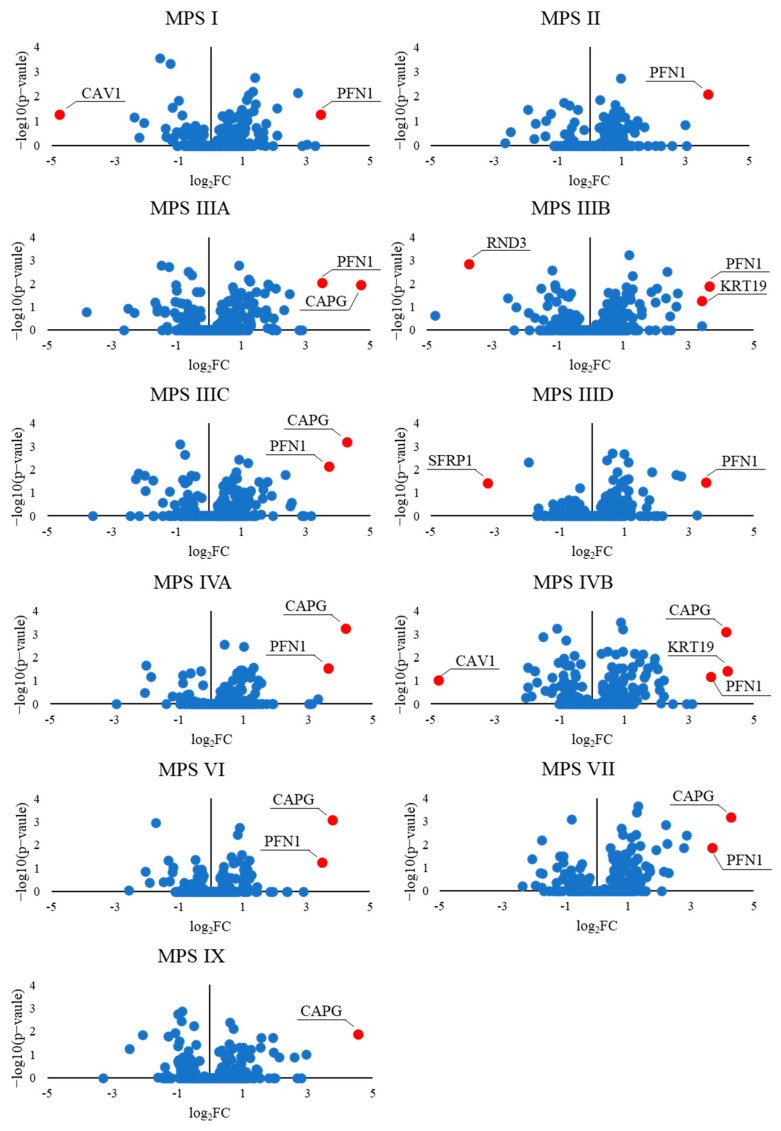
Volcano plots demonstrating an abundance of transcripts of genes included in the actin-filament-based process (GO:0030029) of different MPS types, relative to control cells (HDFa).

**Figure 8 cells-12-01782-f008:**
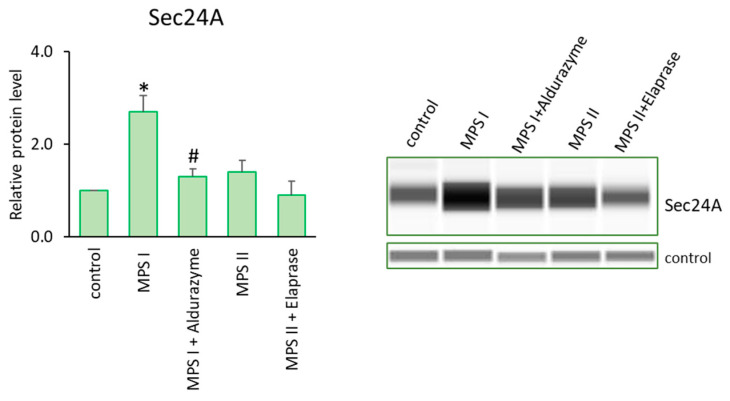
Levels of the Sec24A protein in MPS I and MPS II cells either untreated or treated with the relevant enzyme. Investigated lines of fibroblasts were incubated in the presence of an appropriate enzyme. Relative levels of the proteins were measured using the Western-blotting procedure. The representative blot is shown and data were quantitated via densitometry. Error bars represent standard deviation of three independent repetitions of a given experiment. Differences were considered to be statistically significant relative to the control cell line (*) or untreated MPS I or MPS II (#) when *p* < 0.05.

**Figure 9 cells-12-01782-f009:**
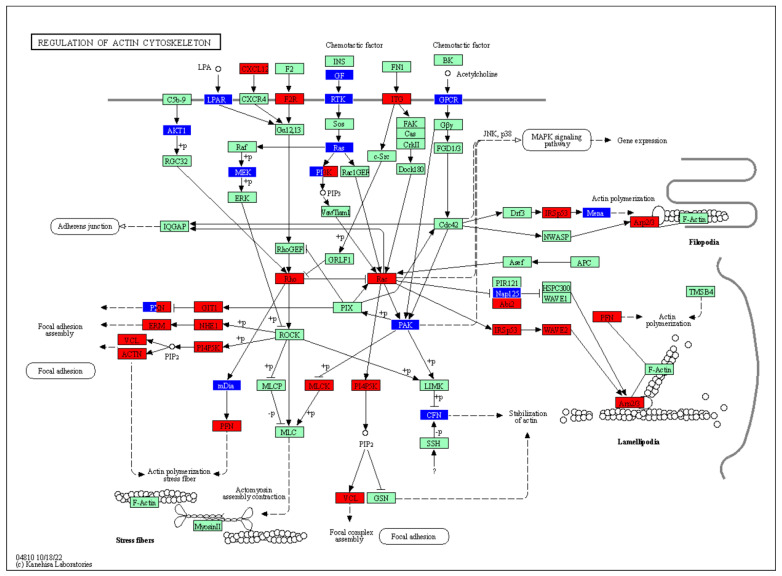
Changes in protein processing in MPS cells. The KEGG pathway presenting ‘Regulation of actin cytoskeleton’ process, imaged from transcriptomic data derived from MPS cells. Individual proteins or groups of proteins were colored if a change in the expression of an indicated gene was observed in at least one type of MPS. Down-regulated and up-regulated genes are marked in blue and red, respectively. Green color indicate results in which no statistically significant differences between MPS and HDFa were determined.

**Table 1 cells-12-01782-t001:** Transcripts with significantly changed levels in MPS fibroblasts relative to control (HDFa) cells (at *p* < 0.1) in at least 5 types of the disease. Up-regulation is marked in red, and down-regulation is marked in blue. Uncolored values indicate no statistically significant differences between MPS and HDFa cells.

Transcript	Log_2_fc Of Selected Transcripts’ Levels In Particular MPS Type Vs Hdfa Line										
	I	II	IIIA	IIIB	IIIC	IIID	IVA	IVB	VI	VII	IX
*BST1*	1.73	1.02	2.07	2.02	0.94	2.59	1.45	1.25	1.45	1.55	0.92
*PFN1*	3.45	3.71	3.51	3.64	3.73	3.53	3.62	3.67	3.50	3.67	2.94
*PDLIM7*	0.42	0.80	0.37	0.73	0.86	0.77	0.81	0.94	0.41	0.78	0.65
*FARP1*	0.58	0.94	0.70	0.41	0.78	0.75	0.78	0.71	0.55	0.57	0.49
*FHL3*	0.72	0.93	0.78	0.50	0.84	0.50	0.92	0.67	0.84	1.38	0.72
*CAMK2D*	1.37	1.23	1.14	0.77	1.30	0.72	1.21	1.56	1.10	0.95	0.95
*CAPG*	1.39	0.91	1.73	0.86	1.55	1.08	1.31	1.61	1.14	2.18	1.92
*CAPG*	1.34	0.89	1.51	0.79	1.45	1.17	1.20	1.45	1.18	2.22	1.94
*CAPG*	2.73	2.98	4.71	1.98	4.27	2.75	4.17	4.14	3.82	4.28	4.55
*PRKCD*	−1.21	−1.12	−1.23	−0.30	−0.74	0.34	−0.65	−0.90	−1.34	−0.88	−0.84
*RND3*	−1.60	−1.41	−1.48	−3.71	−0.67	−1.03	−1.88	−0.52	−1.73	−1.75	−0.99
*RAP2A*	−0.26	−0.50	−0.64	−0.71	−0.55	−0.55	−0.53	−0.56	−0.48	−0.54	−0.86
*PREX1*	−0.52	−0.86	−0.56	−1.32	−1.74	−0.55	−2.02	−1.78	−1.20	−0.88	−2.07

## Data Availability

RNA-seq raw results are deposited in the NCBI Sequence Read Archive (SRA), under accession no. PRJNA562649mRNA. Other raw results are available from the authors upon request.

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
