# Peer review of "Actin Cytoskeleton Polymerization and Focal Adhesion as Important Factors in the Pathomechanism and Potential Targets of Mucopolysaccharidosis Treatment"

_cells, 2023, doi:10.3390/cells12131782_

Round 1
Reviewer 1 Report
Submission ID: cells-2425421
Type: Article
Title: Actin cytoskeleton polymerization and focal adhesion as potential targets for mucopolysaccharidosis treatment
Authors: Lidia Gaffke, Estera Rintz, Karolina Pierzynowska and Grzegorz Węgrzyn
This manuscript describes the analysis of changes in the expression of proteins involved in the actin cytoskeleton and cell adhesion due to mucopolysaccharidosis (MPS). The authors first showed that F-actin is increased in fibroblasts derived from patients with MSPI and MSPII. The authors also analyzed the expression levels of several proteins involved in the actin cytoskeleton in the fibroblasts, as well as in the brain and liver of MPSI model mice. In addition, the authors analyzed the transcriptome analysis in cells with all types of MPS by extracting genes involved in the actin cytoskeleton and cell adhesion, and found genes that have altered expression specifically in MPS. However, the authors only showed data from their analysis, and there were few new findings that could lead to the understanding of the causes of the MPS symptom. And, there is no description of the method of quantification of F-actin shown in Fig. 1 and Fig. 2, and the credibility of the data is suspected.
Major points
1. Introduction
In the Introduction section, from Lines 63 to 101 "In non-neural cells, alternate polymerization ------is one of the hallmarks of neurological diseases.", the authors described the functions of proteins, which are involved in actin cytoskeleton remodeling and cell adhesion. The authors need to describe the relationship of the listed functions to each symptom of MSPs so that the reader can easily understand.
2. Fig. 1 and Fig. 2
In Fig. 1, Stress fibers are observed by staining for F-actin, however, there is no apparent difference between the control, MPS1, and MPSII cells. There is no description of the method of quantification of F-actin at all, so I don't know what the difference is measured. The method of quantification needs to be described.
In Fig. 2, F-actin is increased more than 2-fold in MPSI, II cells compared to normal cells, unlike in Fig. 1, even though the same cells as in Fig. 1 were stained with the same dye.
A 2-fold change in the degree of polymerization of actin indicates that it would be hard to proliferate or to allow the cells to survive for a long period. It is suspected that the measurement method is in error in some way.
To confirm the results of Fig. 1, instead of the analysis in Fig. 2, an experiment should be performed to extract the total actin from the cells, separate G-actin and F-actin by centrifugation, and measure the amount of total actin in the cells and the amount ratio of G- and F-actin by Western blot analysis. The total amount of actin and the G/F ratio in MPS cells should be compared with that in normal cells.
3. discussion
In the Discussion section, from Lines 359 to 409 "When searching for novel therapies for MPS, ------tumors with wild-type p53 protein [62].", the authors provided examples of diseases caused by abnormalities in the actin cytoskeleton and cell adhesion. The authors need to describe the association by stating which of the symptoms caused by the actin cytoskeleton abnormalities listed are similar to those observed in the 11 types of MSP so that the reader can easily understand.
Minor points
1. Fig. 1, 3, 5
The authors showed that Genistein and the active forms of the dysfunctional enzymes, Alurazyme and Elaprase, recover the MPS-induced changes in protein expression, indicating that the changes depend on MPS.
However, it is better to investigate whether the expression of normal Iduronidase and Iduronate sulfatase can recover the changes in their expression in MPSI and MPSII cells.
2.
The authors should indicate to what degree the brains and livers of MSPI mice are or are not injured by MPS.
3.
The proteins altered in MPS cells shown in the results are housekeeping proteins and are unlikely to be targeted for MPS therapy.
To argue "actin cytoskeleton polymerization and focal adhesion could be potential targets for therapies specific to MPS.", the authors should provide additional evidence to suggest the proposal, or to describe previous findings.
Author Response
REVIEWER’S COMMENT:
Major points
- Introduction
In the Introduction section, from Lines 63 to 101 "In non-neural cells, alternate polymerization ------is one of the hallmarks of neurological diseases.", the authors described the functions of proteins, which are involved in actin cytoskeleton remodeling and cell adhesion. The authors need to describe the relationship of the listed functions to each symptom of MSPs so that the reader can easily understand.
RESPONSE: We appreciate this comment of the reviewer. It would be excellent if we could assign specific changes in actin skeleton remodeling and cell adhesion to particular symptoms of MPS. However, at the current stage of our knowledge, it is not possible. In fact, this paper is the fist one demonstrating in a comprehensive way that actin skeleton remodeling and cell adhesion are severely affected in MPS which adds important points to our knowledge on the pathomechanism of MPS. Nevertheless, connection of these changes to particular symptoms of the disease require further intensive studies.
REVIEWER’S COMMENT
- Fig. 1 and Fig. 2
In Fig. 1, Stress fibers are observed by staining for F-actin, however, there is no apparent difference between the control, MPS1, and MPSII cells. There is no description of the method of quantification of F-actin at all, so I don't know what the difference is measured. The method of quantification needs to be described.
RESPONSE: Description of the methods have been improved. Changes introduced during revision are marked by color in the manuscript.
REVIEWER’S COMMENT
In Fig. 2, F-actin is increased more than 2-fold in MPSI, II cells compared to normal cells, unlike in Fig. 1, even though the same cells as in Fig. 1 were stained with the same dye.
A 2-fold change in the degree of polymerization of actin indicates that it would be hard to proliferate or to allow the cells to survive for a long period. It is suspected that the measurement method is in error in some way.
RESPONSE: We assume that quantitative differences can be due to different methods used in experiments shown in Figures 1 and 2. The crucial point is that the direction of the changes and the tendency are the same. Please, see also response to the editor’s comments (below)
REVIEWER’S COMMENTS
To confirm the results of Fig. 1, instead of the analysis in Fig. 2, an experiment should be performed to extract the total actin from the cells, separate G-actin and F-actin by centrifugation, and measure the amount of total actin in the cells and the amount ratio of G- and F-actin by Western blot analysis. The total amount of actin and the G/F ratio in MPS cells should be compared with that in normal cells.
RESPONSE: We appreciate this comment, and the suggested experiments are definitely worth to perform. However, we got only 10 days for the revision, thus, such experiments could be performed during this period. Moreover, the editor did not ask for conducing such experiments. Therefore, we will follow the reviewer’s suggestion in our further work, rather than include them into this paper.
REVIEWER’S COMMENTS
- discussion
In the Discussion section, from Lines 359 to 409 "When searching for novel therapies for MPS, ------tumors with wild-type p53 protein [62].", the authors provided examples of diseases caused by abnormalities in the actin cytoskeleton and cell adhesion. The authors need to describe the association by stating which of the symptoms caused by the actin cytoskeleton abnormalities listed are similar to those observed in the 11 types of MSP so that the reader can easily understand.
RESPONSE: As indicated above (Major points, p.1), the specific connections between actin cytoskeleton and cell adhesion and MPS symptoms are not know, and this work presents for the first time comprehensive studies on changes in actin cytoskeleton and cell adhesion in MPS.
REVIEWER’S COMMENT
Minor points
- Fig. 1, 3, 5
The authors showed that Genistein and the active forms of the dysfunctional enzymes, Alurazyme and Elaprase, recover the MPS-induced changes in protein expression, indicating that the changes depend on MPS.
However, it is better to investigate whether the expression of normal Iduronidase and Iduronate sulfatase can recover the changes in their expression in MPSI and MPSII cells.
RESPONSE: Perhaps our description was not clear enough. As indicated in the manuscript, we have used active (“normal”) forms of Iduronidase and Iduronate sulfatase (as recombinant enzymes, called Alurazyme and Elaprase, respectively). Therefore, the results demonstrate exactly what the reviewer recommended.
REVIEWER’S COMMENT
2.
The authors should indicate to what degree the brains and livers of MSPI mice are or are not injured by MPS.
RESPONSE: The changes in the brains and livers of MPS I mice were reported many times previously which mentioned in the text.
REVIEWER’S COMMENT
3.
The proteins altered in MPS cells shown in the results are housekeeping proteins and are unlikely to be targeted for MPS therapy.
To argue"actin cytoskeleton polymerization and focal adhesion could be potential targets for therapies specific to MPS.", the authors should provide additional evidence to suggest the proposal, or to describe previous findings.
RESPONSE: We would like to emphasize that possible therapies would result in the alleviation of elevated/reduced levels of specific proteins rather than creating total knockouts of the essential housekeeping genes. With that said, we performed an additional experiment (see Fig. 9) and refer to the data we obtained earlier [40]. We have discussed this issue in more detail in Discussion section (lines 439-450).
“This assumption was corroborated through measuring levels of the Sec24A protein (Figure 9). Such a scenario can be also true in the case of genistein treatment. The mechanism of action of genistein, on one hand, is based on the reduction of substrate synthesis, but on the other hand, this compound induces the autophagy process which is currently explored in many diseases as a therapeutic strategy. GAGs play many important functions in cells and tissues, being essential components. Therefore, any lowering of their levels must be balanced. The same should be true for designing new therapeutic targets focused on essential proteins. In fact, genes related to the actin cytoskeleton belong to the housekeeping genes, however, we would like to emphasize that any possible therapies should result in the alleviation of elevated/reduced levels of specific proteins rather than creating genetic knockouts or total silencing of their expression.”
Response To Edtor’s Comments Regarding Recommendations Of The Reviewer no. 1
EDITOR’S COMMENT:
Please revise the manuscript according to comments of reviewers 2, 3. In addition, please refer to comments of reviewer 1 as follows: Please tone down the sentence: "actin cytoskeleton polymerization and focal adhesion could be potential targets for therapies specific to MPS."
RESPONSE:
As recommended, the title and conclusion of the paper have been modified accordingly. We have softened our message. The revised title reads as follows: “Actin cytoskeleton polymerization and focal adhesion as important factors in the pathomechanism and potential targets for mucopolysaccharidosis treatment”. The revised discussion on potential therapeutic targets are included in lines 450-452; 454-458 and read as follow:
“It is possible that a therapy that takes into account the elimination of GAG storage along with effects on additional elements, such as targeting disruption of the actin cytoskeleton, would be the most effective.”
“Expressions of genes coding from proteins related to the actin cytoskeleton and focal adhesion, as well as levels of corresponding protein, are significantly changed in MPS cells and tissues of MPS I mice. These findings support the proposal that actin cytoskeleton polymerization and focal adhesion are both important factors in the pathomechanism and potential subsidiary targets for treatment of MPS.”
EDITOR’S COMMENT:
Please provide additional data showing alterations in 1-2 proteins which are not housekeeping proteins.
RESPONSE:
We would like to emphasize that possible therapies would result in the alleviation of elevated/reduced levels of specific proteins rather than creating total knockouts of the essential housekeeping genes. With that said, we performed an additional experiment (see Fig. 9) and refer to the data we obtained earlier [40]. We have discussed this issue in more detail in Discussion section (lines 439-450).
“This assumption was corroborated through measuring levels of the Sec24A protein (Figure 9). Such a scenario can be also true in the case of genistein treatment. The mechanism of action of genistein, on one hand, is based on the reduction of substrate synthesis, but on the other hand, this compound induces the autophagy process which is currently explored in many diseases as a therapeutic strategy. GAGs play many important functions in cells and tissues, being essential components. Therefore, any lowering of their levels must be balanced. The same should be true for designing new therapeutic targets focused on essential proteins. In fact, genes related to the actin cytoskeleton belong to the housekeeping genes, however, we would like to emphasize that any possible therapies should result in the alleviation of elevated/reduced levels of specific proteins rather than creating genetic knockouts or total silencing of their expression.”
EDITOR’S COMMENT:
Please provide data that the manipulation of the altered proteins improves the symptoms and characteristics of MPS in MSP cells.
RESPONSE:
We are that to demonstrated this, a series of additional experiments would be necessary. However, in our work, we mainly focused on additional factors of MPS pathogenesis and on signaling the possibility of the use of them as potential therapeutic targets in the future. Nevertheless, there are indications derived from the experiments we performed earlier, and to which we refer in the Discussion, that the proposed therapy might indirectly improve the morphology of lysosomes, mitochondria or endoplasmic reticulum. The following text was included (lines 433-440):
„Indeed, as we pointed out earlier, under the influence of the enzyme therapy (for both MPS I and MPS II) there is a reduction in the number of altered lysosomes, which made the cells virtually indistinguishable from healthy ones. Similarly, improvements were obtained in mitochondrial length for MPS II, and in the case of MPS I cells, the treatment with α-L-iduronidase normalized average coverage by endoplasmic reticulum [40]. It turns out that correcting GAG storage with enzymes can improve the levels of proteins that were originally altered. This assumption was corroborated through measuring levels of the Sec24A protein (Figure 9).”
Reviewer 2 Report
The manuscript was reviewed and the comments are as follows.
“Actin cytoskeleton polymerization and focal adhesion as potential targets for MPS treatment” sounds like a promising strategy with potential effects for the treatment of MPS. The main targets of the project should then be aiming at the relatively refractory organs, such as CNS, cornea, cardiac valves, bone and cartilage, etc. to the ERT, HSCT and/or genistein in various types of MPS. Although the authors have provided quite persuasive evidence through meticulous laboratory works, the results were variable sometimes even confusing in different settings. And the main point of this report seemed to suggest genistein could be a good choice of treatment by playing the role of not only inhibiting GAG synthesis, but also modulating actin cytoskeleton and focal adhesion. On this point, more evidence needs to be fortified. It is difficult to work on all different types of mucopolysaccharidoses with so many challenges in the phenotypes and genotypes trying to reach similar and effective “potential targets for treatment”. This report hasn’t yet given the conclusive results. Nevertheless, the authors did provide a new direction of study for improving the treatment outcomes for the MPS patients with or without available clinical therapies.
There are some other minor points for the authors to make corrections or responses:
- In “Abstract”, line 15, Wester-blotting should be Western-blotting.
- In “1. Introduction”, lines 39-40, There are 13 types of mucopolysaccharidoses, classified on….., the authors should provide more supportive evidence against the current knowledge of MPS classification with 11 or 12 types.
- On page 3, “2.3. Reagents”, please clarify if Elaprase was purchased from the no longer existed Shire Human Genetic Therapies.
- On page 11, “3.4 Transcriptomic Analysis of Changes in the Processes Involving Actin Cytoskeleton in MPS Cells”, the results showed, in all cases, there was a preponderance of an increased expression. The authors demonstrated that the highest number of changed transcripts occurred in MPS IVB, while the lowest (expression) occurred in MPS VI. However, the cell lines studied in the project mentioned in “2. Materials and Methods; 2.1 Cell Lines”, only fibroblast lines of MPS I, MPS II and the control were mentioned and clearly described.
- On page 15, the 1st sentence of the “4. Discussion” should be rephrased since it’s a little bit difficult to comprehend.
*Please refer to the “Comments and Suggestions for Authors
”.
Author Response
Reviewer #2:
“Actin cytoskeleton polymerization and focal adhesion as potential targets for MPS treatment” sounds like a promising strategy with potential effects for the treatment of MPS. The main targets of the project should then be aiming at the relatively refractory organs, such as CNS, cornea, cardiac valves, bone and cartilage, etc. to the ERT, HSCT and/or genistein in various types of MPS. Although the authors have provided quite persuasive evidence through meticulous laboratory works, the results were variable sometimes even confusing in different settings. And the main point of this report seemed to suggest genistein could be a good choice of treatment by playing the role of not only inhibiting GAG synthesis, but also modulating actin cytoskeleton and focal adhesion. On this point, more evidence needs to be fortified. It is difficult to work on all different types of mucopolysaccharidoses with so many challenges in the phenotypes and genotypes trying to reach similar and effective “potential targets for treatment”. This report hasn’t yet given the conclusive results. Nevertheless, the authors did provide a new direction of study for improving the treatment outcomes for the MPS patients with or without available clinical therapies.
RESPONSE:
We thank this reviewer for the comment. We were indeed too hasty in introducing the statement about the word "therapies". More specifically, the main goal of our study was to better understand the pathomechanism of MPS, and to signal potential subsidiary targets for a therapy. Therefore, the title of the paper has been modified accordingly, and in the revised version it reads as follows “Actin cytoskeleton polymerization and focal adhesion as important factors in the pathomechanism and potential targets for mucopolysaccharidosis treatment”. Indeed, we have softened our message. We would like to emphasize that possible subsidiary therapies would result in the alleviation of elevated/reduced levels of specific proteins rather than creating total knockouts of the essential housekeeping genes. With that said we have performed an additional experiment (see Fig. 10) and now we refer also to the data we obtained earlier. These aspects are discussed in the revised version in more detail in the Discussion section as follows (lines 433-452):
“Indeed, as we pointed out earlier, under the influence of the enzyme therapy (for both MPS I and MPS II) there is a reduction in the number of altered lysosomes, which made the cells virtually indistinguishable from healthy ones. Similarly, improvements were obtained in mitochondrial length for MPS II, and in the case of MPS I cells, the treatment with α-L-iduronidase normalized average coverage by endoplasmic reticulum [40]. It turns out that correcting GAG storage with enzymes can improve the levels of proteins that were originally altered. This assumption was corroborated through measuring levels of the Sec24A protein (Figure 9). Such a scenario can be also true in the case of genistein treatment. The mechanism of action of genistein, on one hand, is based on the reduction of substrate synthesis, but on the other hand, this compound induces the autophagy process which is currently explored in many diseases as a therapeutic strategy. GAGs play many important functions in cells and tissues, being essential components. Therefore, any lowering of their levels must be balanced. The same should be true for designing new therapeutic targets focused on essential proteins. In fact, genes related to the actin cytoskeleton belong to the housekeeping genes, however, we would like to emphasize that any possible therapies should result in the alleviation of elevated/reduced levels of specific proteins rather than creating genetic knockouts or total si- of their expression. It is possible that a therapy that takes into account the elimination of GAG storage along with effects on additional elements, such as targeting disruption of the actin cytoskeleton, would be the most effective.”
REVIEWER’S COMMENT:
There are some other minor points for the authors to make corrections or responses:
In “Abstract”, line 15, Wester-blotting should be Western-blotting.
RESPONSE:
We thank for noticing this typographical error. It has been corrected.
REVIEWER’S COMMENT:
In “1. Introduction”, lines 39-40, There are 13 types of mucopolysaccharidoses, classified on….., the authors should provide more supportive evidence against the current knowledge of MPS classification with 11 or 12 types.
RESPONSE:
We understand that due to the recently described MPS X (Verheyen et al., 2022) and MPS-plus-syndrome (MPSPS) (Vasilev et al., 2020), the current classification of MPS types may be controversial. Apart from the type that is described only in mice (MPS IIIE), we have distinguished following MPS types: MPS I, MPS II, MPS III A/B/C/D, MPS IV A/B, MPS VI, MPS VII, MPS IX, MPS X, and MPSPS, indicating the total number of 13 types (such a classification has been described recently by WiÅ›niewska, et al. “Misdiagnosis in mucopolysaccharidoses.” Journal of Applied Genetics; 63 (2022): 475-495. doi:10.1007/s13353-022-00703-1).
REVIEWER’S COMMENT:
On page 3, “2.3. Reagents”, please clarify if Elaprase was purchased from the no longer existed Shire Human Genetic Therapies.
RESPONSE:
Once again, we thank the reviewer for catching our error. The Methods section has been modified accordingly. The enzymes were not purchased, but were donated. The correct fragment is now provided in lines 132-136 of the revised manuscript, and says: “Aldurazyme (laronidase, recombinant human α-L-iduronidase; Genzyme, Sanofi Co., Amsterdam, The Netherlands) and Elaprase (idursulfase, recombinant human 2-iduronate sulfatase; Takeda Ltd., Tokyo, Japan) were used at final concentrations 0.58 mg/mL and 0.5 mg/mL, respectively. The enzymes were gifts from the Institute "Mon-ument - Child Health Center” (Warsaw, Poland).”
REVIEWER’S COMMENT:
On page 11, “3.4 Transcriptomic Analysis of Changes in the Processes Involving Actin Cytoskeleton in MPS Cells”, the results showed, in all cases, there was a preponderance of an increased expression. The authors demonstrated that the highest number of changed transcripts occurred in MPS IVB, while the lowest (expression) occurred in MPS VI. However, the cell lines studied in the project mentioned in “2. Materials and Methods; 2.1 Cell Lines”, only fibroblast lines of MPS I, MPS II and the control were mentioned and clearly described.
RESPONSE:
Originally, in Materials and Methods, we included a citation of the paper detailing the transcriptomic studies, because we wanted to avoid repetition. However, to clarify this, we introduced Supplementary Table S1 with all the cell lines used (lines 114-120, Table S1).
“MPS types I (p.Trp402Ter/p.Trp402Ter in the IDUA gene) and II (p.His70ProfsTer29/- in the IDS gene), and the control cell line (control), were employed. All lines were from the Coriell Institute for Medical Research (Camden, NJ, USA) and cultured under standard conditions. Other data for RNA-seq study are deposited with PRJNA562649 (Sequence Read Archive, SRA) number and methods were described previously [17]. A complete list of cells lines used for transcriptomic analysis is shown in Supplementary Table S1.“
REVIEWER’S COMMENT:
On page 15, the 1st sentence of the “4. Discussion” should be rephrased since it’s a little bit difficult to comprehend.
RESPONSE:
In accordance to the reviewer's recommendation, we have changed the initial part of the Discussion. The revised fragment of the text reads as follows: “The main finding described in this report indicate the relevance of actin cytoskeleton elements to the pathomechanism of MPS. Significant changes were observed in expression of genes and levels of proteins involved in the actin cytoskeleton processing and focal adhesion.”
Reviewer 3 Report
Dear Authors,
The presented original work meets all formal requirements and presents a high substantive level of the obtained research results. I highly recommend it for publication. I just have one question: why do the cells shown in Figures 1 and 2 (in both MPS I and MPS II) and treated with the same factors show different morphology and visualization of F-actin using the same cytochemical method? The image of microfilaments in Fig. 2 is not satisfactory, so perhaps a different F-actin visualization technique (e.g. fluorochrome-conjugated phalloidin) should be used to compare these differences. In addition, a clear explanation should be added in the "Results" chapter as to why the morphology of the same cells is so different in Fig. 1 and 2.
Author Response
Reviewer #3:
REVIEWER’S COMMENT:
Dear Authors,
The presented original work meets all formal requirements and presents a high substantive level of the obtained research results. I highly recommend it for publication. I just have one question: why do the cells shown in Figures 1 and 2 (in both MPS I and MPS II) and treated with the same factors show different morphology and visualization of F-actin using the same cytochemical method? The image of microfilaments in Fig. 2 is not satisfactory, so perhaps a different F-actin visualization technique (e.g. fluorochrome-conjugated phalloidin) should be used to compare these differences. In addition, a clear explanation should be added in the "Results" chapter as to why the morphology of the same cells is so different in Fig. 1 and 2.
RESPONSE:
We greatly appreciate the reviewer's comments. As for the observed differences in the cell morphology between Figures 1 and 2, they are due to the need to collect cells for the flow cytometry analysis. We decided to detach them from the plate using trypsin which affected the image and can explain the lack of microfilaments in cells presented in Figure 2, despite using the same dye. As recommended, we have added a corresponding explanatory section in the Results (lines 190-194). We thank for the suggestion to use a different dye; unfortunately the quality of images was comparable in this case due to sample preparation procedures, not the use of an improper one.
“The observed differences in the cell morphology between micrographs in Figures 1 and 2 are due to the need to collect cells for the flow cytometry analysis. The use of trypsin affected the cell shape and it can explain the lack of microfilaments in fibroblasts presented in Figure 2, despite using the same dye as in Figure 1”.
Round 2
Reviewer 1 Report
This manuscript contains a valued analysis of the expression profiles of proteins involved in the actin cytoskeleton and cell adhesion in MPSI and MSPII cells and in the tissues of MPS I model mice. And I admit that the authors toned down their claims and improved on some points. However, the experimental method and its explanation in Fig. 1 and Fig. 2 are completely incorrect. I believe that these issues need to be improved. At the very least, Fig. 2 should be deleted. In addition, contents in the introduction and discussion that have nothing to do with this manuscript have not been revised.
I hope these concerns will be addressed by the authors.
Author Response
RESPONSES TO REVIEWER’S COMMENTS
REVIEWER’S COMMENT:
This manuscript contains a valued analysis of the expression profiles of proteins involved in the actin cytoskeleton and cell adhesion in MPSI and MSPII cells and in the tissues of MPS I model mice. And I admit that the authors toned down their claims and improved on some points.
RESPONSE:
We appreciate this acknowledgement.
REVIEWER’S COMMENT:
However, the experimental method and its explanation in Fig. 1 and Fig. 2 are completely incorrect. I believe that these issues need to be improved.
RESPONSE:
Indeed, we have recognized that description of the quantification of the results presented in former Figures 1 and 2 was accidently omitted in the previous version of the manuscript. We have now included the quantification methods into the Materials and Methods section (marked by color in the revised manuscript), thus, the descriptions are corrected.
REVIEWER’S COMMENT:
At the very least, Fig. 2 should be deleted.
RESPONSE:
According to the reviewer’s recommendation, Figure 2 has been removed from the main text. It was transferred to the supplementary material, as it still contains valuable information which corroborates other results, though we agree that it is a supporting material rather than crucial result, thus, in our opinion it should be optimally demonstrated as a supplementary figure.
REVIEWER’S COMMENT:
In addition, contents in the introduction and discussion that have nothing to do with this manuscript have not been revised.
RESPONSE:
As suggested by the reviewer, the fragments of Introduction and Discussion which are focused on other diseases have been significantly shortened (deletions are indicated in the revised manuscript as ‘tracking changes’). We still believe that comparison of the roles of the actin cytoskeleton in different neurodegenerative diseases is very important for this paper, but we agree that in the previous versions these descriptions were too long. Thus, they have been shortened considerably in the revised manuscript.
REVIEWER’S COMMENT:
I hope these concerns will be addressed by the authors.
RESPONSE:
As requested by the reviewer, all points indicated in the second review have been addressed, as described above.